# Supramolecular Chromatographic Separation of C_60_ and C_70_ Fullerenes: Flash Column Chromatography vs. High Pressure Liquid Chromatography

**DOI:** 10.3390/ijms22115726

**Published:** 2021-05-27

**Authors:** Subbareddy Mekapothula, A. D. Dinga Wonanke, Matthew A. Addicoat, David J. Boocock, John D. Wallis, Gareth W. V. Cave

**Affiliations:** 1School of Science and Technology, Nottingham Trent University, Clifton Lane, Nottingham NG11 8NS, UK; subba.mekapothula@ntu.ac.uk (S.M.); Dinga.Wonanke@ntu.ac.uk (A.D.D.W.); matthew.addicoat@ntu.ac.uk (M.A.A.); john.wallis@ntu.ac.uk (J.D.W.); 2The John van Geest Cancer Research Centre, Nottingham Trent University, Clifton Lane, Nottingham NG11 8NS, UK; david.boocock@ntu.ac.uk

**Keywords:** pyrogallol[4]arene, flash column chromatography, HPLC, stationary phase, fullerenes, size-selective molecular recognition

## Abstract

A silica-bound *C*-butylpyrogallol[4]arene chromatographic stationary phase was prepared and characterised by thermogravimetric analysis, scanning electron microscopy, NMR and mass spectrometry. The chromatographic performance was investigated by using C_60_ and C_70_ fullerenes in reverse phase mode via flash column and high-pressure liquid chromatography (HPLC). The resulting new stationary phase was observed to demonstrate size-selective molecular recognition as postulated from our in-silico studies. The silica-bound *C*-butylpyrogallol[4]arene flash and HPLC stationary phases were able to separate a C_60_- and C_70_-fullerene mixture more effectively than an RP-C_18_ stationary phase. The presence of toluene in the mobile phase plays a significant role in achieving symmetrical peaks in flash column chromatography.

## 1. Introduction

Since the discovery of fullerenes in 1985 [1,2], there has been an ever increasing scientific and industrial interest in fullerenes due to their remarkable electronic and optoelectronic properties, charge transfer ability, efficient singlet oxygen sensitising ability, strong electron acceptor character, and superconductivity upon doping with alkali metals. These properties make fullerenes and their derivatives particularly useful in the development of biosensors [3,4], skin preparation and cosmetics [5,6], and a range of other medical applications [7,8]. Fullerenes-based nanomaterials have great potential for a plethora of biomedical applications but are limited by the high cost and difficulties in purification. This difficulty is a result of the structural similarities, specifically the exclusive sp^2^ carbon atoms of the hollow ball-shaped fullerenes [9].

The most commonly used techniques for the separation and purification of fullerenes are crystallisation [10], complexation [11], and chromatographic techniques [12]. Each method has its advantages and disadvantages; for example, complexation methods require the use of pre-engineered molecules, which are expensive and difficult to recycle. Crystallisation, on the other hand, is time consuming and with no facile means for automation.

Many novel liquid chromatographic stationary phases have been designed and synthesised for effective separation and isolation of fullerenes, such as activated carbon for separation of C_60_/C_70_ fullerene mixtures [13] and metal organic frameworks (MOFs) for separation of a mixture of eight different fullerenes from C_60_ up to C_84_ [14]. There are many reverse-phase stationary phases available for the separation of pristine fullerenes (C_60_, C_70_, C_76_, C_78_, C_82_ and C_84_), i.e., silica functionalised with C_18_ chains [15], humic acids [16], tetraphenylporhyrin [17], metal-organic frameworks (MOFs) [18] and pyrenebutyric acid [19]. Although columns with these stationary phase materials are available commercially, optimisation is carried out empirically and allows only for qualitative separation of fullerenes [20]. The advancement in the LC separation and purification of fullerenes is typically stationary phase dependent. Consequently, several studies have investigated the use of specific selectors, with one or multiple interaction mechanisms with fullerene molecules, as stationary phases for the LC separation of fullerite [13,14,15,16,17,18,19].

The design of new molecules as specific selectors to provide host–guest systems mediated through non-covalent interactions is one of the vital goals of supramolecular chemistry [21,22]. Calixarenes are referred to as third generation supramolecules, discovered after crown ethers and cyclodextrins. They possess cavities and exhibit versatile complexation abilities with guest molecules [23,24]. Calixarenes’ guest recognition and their ability to form a reversible complex with neutral as well as charged molecules make them excellent tools in analytical chemistry [25,26,27]. There have been preliminary attempts for purification and separation of fullerenes using supramolecular cavitands via cyclodextrins [28,29], calixarene macrocycles [30] and porphyrin nano-barrels [31]. Selective extraction and precipitation of C_70_ over C_60_ was achieved using *p*-halohomooxacalix[3]arenes and calix[5]arene [32]. The drawback of this approach is the significant difficulty encountered in the release and recovery of valuable fullerenes from the complex. A variety of host–guest interaction mechanisms, including hydrophobic, π–π interactions, hydrogen bonding and inclusion interactions, were exploited in the separation process. It was observed that supramolecular chromatographic stationary phases provide multiple mode of interactions in comparison to conventional normal or reversed phase chromatographic stationary phases.

In this paper, we describe the synthesis of the supramolecular cavitand *C*-bromo-butylpyrogallol[4]arene **2** from *C*-hydroxybutylpyrogallol[4]arene **1,** which itself is synthesised via previously reported microwave irradiation in high yields [33]. Subsequently, *C*-bromobutylpyrogallol[4]arene **2** was covalently linked to silica gel particles. The new stationary phase **3** was evaluated to study the chromatographic behaviour of C_60_ and C_70_-fullerenes using flash column chromatography and high-pressure liquid chromatography (HPLC) in comparison with a reverse phase C_18_ stationary phase. Moreover, an in-silico study was performed to further gauge the host–guest interactions between the butylpyrogallol[4]arene stationary phase **3** and the two fullerenes.

## 2. Results and Discussion

*C-*Alkylpyrogallol[4]arenes (PgCn) are a subgroup of calixarene macrocycles, which share two key features. The first is they comprise twelve hydroxy groups at the upper rim of their cup-like structure, which results in many hydrogen-bonding interactions. Second is the alkyl groups that radiate from the spheroidal assembly, which range in length from ethyl to undecyl [34,35]. This abundance of potential hydrogen bonding sites led to the synthesis of a number of pyrogallol[n]arene derivatives [36,37] to study not only chromatographic applications of pyrogallol[n]arenes but also application in the field of separation science.

### 2.1. Synthesis of Silica-Bound C-Butylpyrogallol[4]arene Stationary Phase ***3a*** and ***3b***

*C*-Hydroxybutylpyrogallo[4]arene **1** was prepared from pyrogallol and 2,3-dihydropyran under microwave irradiation [33] and converted to the tetrabromo derivative **2** with phosphorus tribromide in 64% yield (Scheme 1). The ^1^H NMR for *C*-bromobutylpyrogallol[4]arene **2** revealed the presence of two singlet signals for hydroxy protons in the downfield region, as well as multiplet signals for the bridge proton and the methylene protons adjacent to bromine. The multiplet signals for the rest of the aliphatic side chain protons are in good agreement with the proposed structure (Figure 1). The mass spectrometric analysis (ESI-MS) of **2** confirmed the presence of the molecular ion as five m/z peaks, [M]^+^ to [M+8]^+^, in an appropriate ratio for a tetrabromo compound (Appendix A). The silica-bound *C*-butylpyrogallol[4]arene HPLC stationary phase is reported as phase **3a**, while the silica-bound C-butylpyrogallol[4]arene flash stationary phase is reported as phase **3b.**

*C*-Bromobutylpyrogallol[4]arene **2** was attached to the surface of chromatographic silicas (HPLC or flash) using the previously reported method, as shown in Scheme 1 [21,22]. The resultant chromatographic stationary phases **3a** and **3b** were washed with methanol: acetone (70:30) to remove any unreacted pyrogallol[4]arene cavitand and dried. The specifications for these functionalised silicas for HPLC and flash column chromatography are reported in Table 1.

### 2.2. Characterisation of the Silica-Bound C-Butylpyrogallol[4]arene Stationary Phases ***3a*** and ***3b***

The silica-bound *C*-butylpyrogallol[4]arene stationary phases **3a** and **3b** were characterised by thermogravimetric analysis (TGA) to determine the loading of macrocyclic cavitands and by scanning electron microscopy (SEM) to compare the structural morphology of uncoated silica and *C*-butylpyrogallol[4]arene-bound silicas. The thermogravimetric analysis studies were carried out over a temperature range of 25 to 900 °C at 10 °C per minute, as shown in Figure 2. The TGA results confirmed the mass loading of the *C*-butylpyrogallol[4]arene on the surface of the functionalised silica gel at 17.5% (0.01613 M) for the HPLC phase **3a** and at 14.1% (0.00792 M) for the flash column chromatography phase **3b** (Figure 2a,b).

The structural integrities of the silicas modified with *C*-butylpyrogallol[4]arene for HPLC and flash chromatography, **3a** and **3b**, were assessed via SEM and demonstrated to be comparable to the starting spherical silica particles, as shown in Figure 3. They were observed to have smooth surfaces without any cracks or well-defined cavities, which are vital for free flow of the mobile phase and analytes [38,39,40,41]. Consequently, we observed a notably low back pressure during the conditioning of the column and chromatographic evaluation processes.

### 2.3. Supramolecular Chromatographic Separation of C_60_ and C_70_ Fullerenes

The HPLC column packing material **3a** was packed into a stainless-steel HPLC column by a wet slurry packing method (150 × 4.6 mm i.d.) using methanol as the displacing agent at a constant pressure of 34.5 MPa. The flash column stationary phase material **3b** was dry-packed into a flash purification cartridge (12 gm, 62 mm × 12 mm i.d.).

#### 2.3.1. HPLC Separation of C_60_ and C_70_-Fullerenes

Chromatographic methods are currently satisfactory for small-scale purification of not only C_60_ and higher fullerenes from fullerene soot but also for the purification of modified fullerene derivatives. Liquid chromatography (LC) is the major chromatographic technique available for separation of fullerenes due to advances in modern LC chromatographic techniques and instrumental developments, which are readily available to separate fullerenes at the analytical scale.

Since the discovery of fullerenes, to enable these new developments, several *p-*butyl pyrogallol[4]arenes have been extensively investigated for the isolation of fullerenes via host–guest complexation, such as *p*-butyl-calix[8]arene [30], 5-nitro-11,17,23,29-tetramethylcalix[5]arene [42] and *p*-tert-butylcalix[6]arene bearing one *O*-butanoic acid side chain [43]. A number of chromatographic phases have been reported to separate fullerene mixtures include Lewis base modified magnesia–zirconia [44] monomeric type C_30_ stationary phase [45], ligand-modified silica stationary phases, such as *p*-nitrobenzoic acid and naphthyl-acetic acid silica [46]; a core–shell biphenyl stationary phase [47], pyrenebutyric acid bonded silica [48], porphyrins immobilised silica [49], heavy atom containing silica [50] and Phenomenex Cosmosil Buckyprep [51].

However, these calixarene cavitand-bound stationary phases and other modified LC stationary phases have limited use in routine isolation and purification of fullerene mixture due to the requirements of a column with rapid analysis and increased sample loading capacity. It is also important to have green methods for the preparation of macrocycles for use in the separation of fullerenes [33].

The newly synthesised HPLC-silica-bonded *C*-butylpyrogallol[4]arene stationary phase **3a** was primed with water:methanol (1:1) at 0.1 mL/min overnight to condition the column for further chromatographic studies. C_60_ and C_70_ fullerenes were used as probes for investigation of the chromatographic performance of the silica-bonded *C*-butylpyrogallol[4]arene HPLC stationary phase **3a**, and their chromatographic performance was compared with a reverse phase C_18_ column, as shown in Figure 4, using reverse phase mode conditions utilising a mobile phase gradient. As shown in Figure 4a, a 2 µL injection volume of 50 µg of C_60_ and C_70_ fullerenes dissolved in toluene (1 mL) was separated on a traditional RP-C_18_ column via isocratic elution using a modified mobile phase (water:methanol, 40:60) at a flow rate of 1.0 mL min^−1^ using a photodiode array (PDA) detector at a wavelength of 237 nm [52]. These chromatographic conditions resulted in the elution of C_60_ and C_70_ fullerenes at 2.52 and 3.35 min on the RP-C_18_ column, as shown in Figure 4a.

Subsequently, the same chromatographic conditions were utilised to demonstrate the separation of C_60_ and C_70_ fullerenes on the silica-bonded *C*-butylpyrogallol[4]arene HPLC stationary phase **3a**, and resulted in the prominent base-line separation of C_60_-fullerene at 1.08 min and C_70_-fullerene at 4.85 min under 10 min, as shown in Figure 4b. The chromatographic results confirm the better separation and resolution of C_60_ and C_70_ fullerenes on the silica-bonded *C*-butylpyrogallol[4]arene HPLC stationary phase **3a** compared to using an RP-C_18_ HPLC column under identical chromatographic conditions.

#### 2.3.2. Flash Column Separation of C_60_ and C_70_-Fullerenes

While HPLC is the most effective and powerful chromatographic technique for fullerene purification at the analytical scale, it is impractical for large purification of fullerenes due to its high cost, solvent incompatibility in reverse phase stationary phases, low output quantity, limit in column loadings and injection volumes of the order of 1 mL.

Technical and instrumental advances in flash column chromatography and HPLC has allowed the implementation of a single compact system with a flash column chromatography environment and preparative HPLC pumps, which offers a quaternary gradient and a dedicated purge line. It is possible to achieve large scale purification of fullerenes along with the use of a widely adopted LuerLock series of flash columns capable of withstanding up to 22 bar pressure and flow rates ranging from 4 to 300 mL/min, stainless steel HPLC columns for scouter sample analysis and preparative stainless-steel columns for high-pressure applications [53].

However, flash column chromatography is still limited for the effective and rapid preparative separation of fullerenes due to the requirement of a molecular recognition selector capable of making strong enough interactions with fullerenes. Typical normal phase silica gel interacts with fullerenes via van der Waals interactions resulting in unresolved fullerenes with quick elution even with use of a weak nonpolar solvent like hexane. The same issues arise with reverse phase silica gel.

In order to develop a flash column chromatographic stationary phase with novel selectors, Schipper et al. for the first time reported a silica stationary phase consisting of functionalised bent aromatic molecules (iptycenes). This was shown to be suitable for scalable purification of fullerenes by flash chromatography using a UV detector. In their work, 25 mg of a mixture containing fullerenes C_60_ and C_70_ was loaded on the silica functionalised with bent aromatic molecules (12 gm), using hexane as a mobile phase. This resulted in the elution of C_60_ at 45 min and C_70_ at 90 min. This clearly shows the bent aromatic molecules provide a large surface area for increased π–π interactions resulting in high resolution and retention times. However, when the mobile phase was changed to hexane: toluene (3:1), there was a reduction in elution times of C_60_ and C_70_ to 33 and 59 min. Even though there is significant separation and resolution of fullerenes on this stationary phase at milligram levels, the fullerenes eluted with long retention times, and also some loss of functionalised cavitand has been observed [54].

The flash column chromatographic separation of a mixture of fullerene C_60_ and C_70_ was performed on the silica-bound *C*-butylpyrogallol[4]arene stationary phase **3b**. As the chromatographic separation of fullerenes is proportional to the solubility of fullerenes, 25 mg mL^−1^ of each of the C_60_ and C_70_ fullerene standards were dissolved in toluene, and the flash column chromatographic separation of fullerenes on silica-bound *C*-butylpyrogallol[4]arene stationary phase **3b** was carried out using ethyl acetate:toluene (35:65) as a mobile phase. As fullerenes are freely soluble in toluene, a mobile phase combination of ethyl acetate:toluene allows the fullerenes to move freely through the pores of silica-bound C-butylpyrogallol[4]arene stationary phase **3b**. This allows a ball and socket type of reversible host–guest size-selective molecular recognition between the upper aromatic rings of the pyrogallol[4]arene cavity and fullerenes. Subsequently, the stationary phase **3b** resulted in superior and rapid separation of fullerenes C_60_ and C_70_, in 3.20 min and 10.30 min, as shown in Figure 5a,b.

In order to validate the chromatographic performance of the silica-bound *C*-butylpyrogallol[4]arene stationary phase **3b**, we have studied the separation of mixtures of C_60_ and C_70_-fullerenes using the same chromatographic conditions. The silica-bound *C*-butylpyrogallol[4]arene stationary phase **3b** separated out the mixture into two discreet bands at the same retention times, as shown in Figure 5c.

Under these chromatographic conditions, the separation mechanism can be classified as a traditional or normal phase separation where the stationary phase is polar and the mobile phase is nonpolar. The silica-bound *C*-butylpyrogallol[4]arene flash stationary phase **3b** is able to separate the same amount (25 mg) of fullerenes C_60_ and C_70_ and reduce the retention times from 30 to 90 min to 3–11 min compared to the phase of Schipper et al. This shows the rapid separation ability of this silica-bound C-butylpyrogallol[4]arene stationary phase for the separation of a mixture of fullerenes C_60_ and C_70_ with minimal use of solvent.

### 2.4. Quantum Chemistry Calculations

Our in-silico host–guest interaction study was performed in order to further our understanding of the remarkable molecular recognition that results in the favourable chromatographic separation of C_60_ and C_70_ using the *C*-butylpyrogallol[4]arene stationary phases **3a** and **3b**. The Gibbs free binding energies of the gas and solvent phases for the most strongly binding motifs are presented in Table 2 alongside their chromatographic retention time. Accompanying structures are presented in Figure 6. All calculations were undertaken using the DFTB engine in AMS and employed the mio-1-1 parameter set and UFF dispersion [55,56,57,58,59,60].

The Gibbs free binding energies presented in Table 2 show stronger interactions of C_70_ with the cavity in comparison to C_60_. It is important to note that each of these binding motifs are the actual local minima of the potential energy surface, with zero imaginary frequencies. The stronger interaction of the cavity with C_70_ is in line with experimental chromatographic separation, wherein C_70_ shows a remarkably higher retention time both in HPLC and flash column chromatography. The interaction of the fullerenes and the cavity is held in place by a series of strong non-covalent interactions; however, these interactions are similar for all fullerenes and consequently cannot be used as a proxy for explaining this favourable chromatographic separation. Rather, we posit that this preferential interaction of the cavity with C_70_ in comparison to C_60_ can most likely be attributed to size-selective molecular recognition. To test this hypothesis, we computed additional interaction energies and binding motifs for C_50_⊂butylpyrogallol[4]arene and C_80_⊂butylpyrogallol[4]arene for which their solvated Gibbs free binding energies were −123.791 kcal mol^−1^ and −125.566 kcal mol^−1^, respectively. This result shows that the interaction of the fullerenes with the cavity increases with the size of the fullerene structure from which we can develop the following interaction strengths between the fullerene and the cavity: C_80_ > C_70_ > C_60_ > C_50_. Consequently, the *C*-butylpyrogallol[4]arene cavity seems to be an effective stationary phase for purification of fullerenes and should be trialled for other size-selective chromatographic separations.

Moreover, Gibbs free binding energies were also computed for C_60_⊂RP-C_18_ and C_70_⊂RP-C_18_ in order to compare between the *C*-butylpyrogallol[4]arene cavity and RP-C_18_ as stationary phases for the selective separation of fullerenes. The details of these computations are found in the electronic supporting information (ESI). The change in Gibbs free binding energies presented in Appendix A shows stronger interactions of C_70_ with RP- C_18_ in comparison to C_60_. However, the fullerene⊂RP-C_18_ interactions are significantly thermodynamically unfavourable. Consequently, C-butylpyrogallol[4]arene, which has a favourable thermodynamics interaction with fullerenes, is a more suitable stationary phase for the selective separation of fullerenes.

## 3. Materials and Methods

### 3.1. Reagents and Instruments

All chemicals and solvents were purchased as reagent grade or LC-MS grade used without further purification. Pyrogallol, 2,3-dihydropyran, p-toluenesulphonic acid and phosphorous tribromide were purchased from Sigma Aldrich. Microwave synthesis was carried out using a Biotage^®^ microwave reactor via sealed microwave reaction vials. Reactions were monitored by TLC plates (pre-coated with 60 Å silica gel, F254) purchased from Merck KGaA and visualised by UV light (254, 365 nm) or iodine. Flash column chromatography was performed using silica gel (silica flash P60 from Fluorochem Ltd, Glossop, UK) as the stationary phase for the purification of synthetic compounds. ^1^H and ^13^C NMR spectra were recorded on a Jeol 400 MHz NMR ECX-400 spectrometer (Welwyn Garden City, UK) at 25 °C, with chemical shift measured in ppm and referenced by the residual undeuterated solvent peak. Mass spectrometric analysis of compounds was performed using a Waters MS Xevo G2-XS qTOF (Hertfordshire, UK)through direct infusion spray. A Perkin Elmer thermogravimetric analyser TGA4000 (Bucks, UK) was used for thermogravimetric analysis. A Jeol JSM-7100F Field Emission Scanning Electron Microscope (Welwyn Garden City, UK). was used for observing the structural integrity of the new supramolecular stationary phase. Flash column chromatography was carried out using an automated PuriFlash^®^5.125 (Chester, UK) flash system with pre-packed and dry self-packed flash cartridges. Interchim^®^ (Chester, UK) flash (12 g) cartridges packed with 50 μm silica particles with a surface area of 500 m^2^/g were used for flash column chromatographic studies. HPLC grade silica (5 μm) with a surface area of 120 Å was purchased from Alfa Aesar Ltd (Heysham, UK). to develop a new HPLC stationary phase.

### 3.2. Synthesis of C-Hydroxybutyl-Pyrogallol[4]arene ***1***

*C*-Hydroxybutylpyrogallol[4]arene **1** was synthesised via the previously reported one-pot microwave-assisted irradiation reaction [33]. 2,3-Dihydro-pyran (0.912 mL, 0.01 moles) was added to a solution of pyrogallol (1.26 g, 0.01 moles) and p-toluenesulphonic acid (0.1g, 0.58 mmol) in ethanol (ca. 10 mL) contained in a sealed CEM pressure vial. The mixture was pre-stirred (30 s) and heated (100 °C) in a Biotage Initiator 60 instrument by microwave irradiation (130 W, 2 min. power cycle, total of 12 min.). Upon cooling to room temperature, the precipitate was collected by vacuum filtration and washed successively with an ethanol and water mixture (4:1, 5 × 60 mL). This yielded the title compound as a white solid in 92% yield (1.90 g, 3.70 mmol). ^1^H NMR (400 MHz, d_6_-Acetone) δ = 8.74 (br. s, 8H, OH), 8.18 (s, 4H, OH), 6.78 (s, 4H, Ar-H), 4.71 (s, 4H, (CH_2_)OH), 4.21 (t, ^3^J = 7.8 Hz, 4H, CHAr_2_), 3.61 (t, ^3^J = 7.8 Hz, 8H, CH_2_OH), 1.89 (m, 8H, CH_2_), 1.44 (m, 8H, CH_2_), 1.18 (m, 8H, CH_2_) ppm. ^13^C NMR (101 MHz, d_6_-Acetone) δ = 139.5, 132.8 124.5, 113.5, 61.0, 39.1, 35.1, 31.7, 24.3 ppm. MS -TOF *m*/*z* for C_44_H_56_O_16_ calculated 840.36, found 840.3612 [M]^+^.

### 3.3. Synthesis of C-Bromobutylcalix[4]pyrogallolarene ***2***

*C-*4-Hydroxybutyl-pyrogallol[4]arene **1** (4.2 g, 5 mmol) was dissolved in DCM (30 mL) and cooled down at 0 °C under nitrogen. Phosphorous tribromide (2.84 mL, 30 mmol, 6.0 equiv) was added dropwise over 10 min and the mixture was stirred until it reached room temperature (1.5 h). The reaction mixture was stirred at 40 °C for 5 h. The reaction mixture was then evaporated under vacuum, and the resulting oil was sonicated in water (30 mL) for 15 min, filtered and dried under vacuum to obtain a pink solid (3.50 g 3.2 mmol, 64%).^1^H NMR (400 MHz, d_6_-Acetone) δ = 8.64 (s, 8H, OH), 8.08 (s, 4H, OH), 6.88 (s, 4H, Ar-H, 1H), 4.13 (t, ^3^J= 7.8 Hz 4H, 7-H), 3.52 (t, ^3^J = 7.8 Hz 8H, 11-H_2_), 1.84 (m, 8H, 10-H_2_), 1.43 (m, 8H, 8-H_2_) 1.20 (m, 8H, 9-H_2_) ppm. ^13^C NMR (101 MHz, d_6_-Acetone) δ = 145.1, 137.1, 127.2, 119.5, 38.2, 35.2, 33.6, 32.8, 24.8 ppm. ESI-MS *m*/*z* calculated for C_44_H_52_^79^Br_4_O_12_ was 1088.01, found: 1088.0192, 1090.0216, 1092.0201, 1094.0112 and 1096.0109 [M]^+^. (^13^C NMR and mass spectra are in the ESI).

### 3.4. Preparation of C-Butylpyrogallol[4]arene Bound Silica Gel HPLC Stationary Phase ***3a***

HPLC grade silica (1 g, AlfaAesar, CAS: 7631-86-9) was stirred overnight at room temperature in a mixture of THF and TEA (1:1, 50 mL). The solvents were evaporated, and the solid residue was dried at room temperature. *C*-Bromobutyl-calix[4]pyrogallolarene **2** (1 g, 0.915 mmoles) was dissolved in acetone and added to the dried silica in a round-bottomed flask and stirred overnight at room temperature. The acetone was evaporated and the solid product was dried.

### 3.5. Preparation of C-Butylpyrogallol[4]arene Bound Silica Gel Flash Column Stationary Phase ***3b***

Silica (12 g Interchim^®^ (Chester, UK) flash) was stirred overnight at room temperature in a mixture of THF and TEA (1:1, 50 mL). The solvent mixture was evaporated, and the solid residue was dried at room temperature. *C*-Bromobutyl-calix[4]pyrogallolarene **2** (1 g, 0.915 mmoles) was dissolved in acetone (50 mL) and added to the previously dried silica in a round-bottomed flask and the mixture stirred overnight at room temperature. The acetone was evaporated and the solid product was dried and dry-packed into a flash purification cartridge.

### 3.6. Column Packing

The silica-bound *C*-butylpyrogallol[4]arene **3a** was packed into HPLC stainless steel tubes (150 × 4.6 mm i.d.) by a slurry packing method under a constant packing pressure of 40 MPa, using methanol as both suspension solvent and propulsion solvent. Silica-bound *C*-butylpyrogallol[4]arene **3b** and was dry-packed by tapping into an empty Interchim^®^ (Chester, UK) flash column cartridge (62 × 12 mm i.d.) for flash column chromatography and subsequently primed with methanol: ethyl acetate (4:1) to remove unreacted materials.

### 3.7. HPLC Separation Procedure

A *C*-butylpyrogallol[4]arene bonded-silica stationary phase column **3a** was equilibrated with methanol:water (80:20). All chromatographic separations were performed on a PerkinElmer Flexar UHPLC system (Bucks, England), consisting of a Flexar fx-15 UHPLC pump, an autosampler injector equipped with 20 μL PEEK sample loop and a Flexar PDA UV detector. Separations were performed at a temperature of 25 °C, a flow rate of 1.0 mL/min, and using a detection wavelength of 237 nm. In order to ensure the reliability of the obtained data, all the separation experiments were repeated five times.

### 3.8. Flash Column Chromatographic Procedure

A *C*-butylpyrogallol[4]arene bonded-silica stationary phase column **3b** was equilibrated by in-built optimisation conditions. A total of 50 mg of a mixture of C_60_ and C_70_ fullerenes was dissolved in toluene (2 mL). The mobile phase used for the analysis was ethyl acetate:toluene (35:65) in isocratic conditions. The flow rate was set to 15 mL/min, and UV detection was set to 200–400 nm. Flash column chromatography was carried out via an automated PuriFlash^®^5.125 (Chester, UK) flash system. The column was first equilibrated by running through 15 mL of elution solvent. A mixture of analytes in toluene was loaded into the column. Fraction collection volume was set to 13 mL.

### 3.9. In Silico Study for the Host–Guest Interaction

To gauge the most likely binding motifs and energies between the fullerenes and *C*-butylpyrogallol[4]arene, 100 fullerenes⊂*C*-butylpyrogallol[4]arene complexes were generated for each fullerene (C_50_, C_60_, C_70_ and C_80_) via a semi-stochastic method in order to compute their host–guest interactions. In this semi-stochastic approach, a random structure generation algorithm known as Kick, which was developed by one of the authors, was used [55]. In this method, specified molecular fragments are randomly translated in a chosen virtual box providing chemically sensible translated entities. For this study, the *C*-butylpyrogallol[4]arene cavity was considered to be the centre of the virtual box with a predefined radius of 2.8 Å, which is the radius of the cavity. The fullerenes were then randomly translated into the box at different spatial positions. This was supplemented by a heuristic approach, where the coordinates of the fullerene were manually translated along the interior of the *C*-butylpyrogallol[4]arene cavity, using the Gaussview 6.0 visualiser [56]. In this process, new coordinates were obtained by inserting all regions of the optimised fullerene into the *C*-butylpyrogallol[4]arene cavity. This was done in order to ensure a broad sampling of possible fullerene⊂*C*-butylpyrogallol[4]arene complexes.

Once the 100 complexes for each fullerene were generated, a geometry optimisation was performed for all these structures using density functional tight binding (DFTB) with the mio-1-1 [57] parameter set including the universal force field, UFF, [58] with dispersion correction UFF-DFTB/mio-1-1 as implemented in the Amsterdam Modelling Suite AMS version adf2019.305 [59]. We then selected 20 energy minima of each of these complexes and performed a vibration frequency calculation in the gas phase in order to verify that they were actual local minima on the potential energy surfaces. Once confirmed that these structures were all local minima, we performed a geometry and frequency optimisation in solvent (toluene) using the implicit generalised Born solvation model with solvent accessible surface area (GBSA) at the same level of theory as in the gas phase [60]. A total of 2030 surface grid points were used in order to ensure smooth solvent phase geometry optimisation with as little numerical noise as possible. All the structures, long range interactions and energies of these 20 energy minima can be freely accessed at http://doi.org/10.5281/zenodo.4147306 (last accessed on 21 May 2021). 

The geometry optimised gas and solvent phase Gibbs free binding energies (ΔGBE), were then computed for these 20 energy minima with the formula given in Equation (1):ΔGBE = G_complex − (G_fullerene + G_cavity)(1)
where G_complex represents the Gibbs free energy of the fullerene⊂*C*-butylpyrogallol[4]arene complex; G_fullerene and G_cavity represent the Gibbs free energy of the lowest energy (geometry optimised) isolated fullerene and the *C*-butylpyrogallol[4]arene cavity, respectively, i.e., the computed ΔGBE contain the energy contribution from deforming both the fullerene and the cavity.

## 4. Conclusions

In conclusion, silica-bound *C*-butylpyrogallol[4]arene **3** can be effectively synthesised and has demonstrated superior separation resolution of C_70_-fullerene compared to C_60_-fullerene via size-selective molecular recognition using flash column chromatography and HPLC. The flash column pyrogallol[4]arene stationary phase **3b** showed a significant increase of isolated quantities of fullerenes to milligrams under 15 min in comparison to micrograms separation from the HPLC supramolecular stationary phase **3a**. The flash column and HPLC grade silica-bound *C*-butylpyrogallol[4]arene phase showed superior separation selectivity via size-selective molecular recognition than an RP-C_18_ column separation selectivity for the mixture of C_60_ and C_70_-fullerenes.

## Data Availability

Data available at libinfodirect@ntu.ac.uk.

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
