# Peer review of "Supramolecular Chromatographic Separation of C60 and C70 Fullerenes: Flash Column Chromatography vs. High Pressure Liquid Chromatography"

_ijms, 2021, doi:10.3390/ijms22115726_

Round 1

Reviewer 1 Report

Supramolecular host-guest complexation has the potential to afford efficient large-scale purification of organic molecules. The manuscript reports a systematic study on the separation of C60 and C70 fullerenes using a silica‐modified C‐butylpyrogallol[4]arene chromatographic stationary phase applied in flash column chromatography and high pressure liquid chromatography in a reverse phase mode. The prepared materials showed excellent separation efficiency based on the different affinity of the C60 and C70 to the macrocyclic cavity. The manuscript is well-written and the results of significant importance. Therefore, I recommend the publication of the manuscript after considering the following points:

  • Could the authors comment on the smaller size of modified silica particles compared to the unmodified ones, as shown in the SEM images, Figure 3.
  • In the TGA analysis of the modified silica, Figure 2, the Cbutylpyrogallol[4]arene loading on the surface was determined by weight loss, however, the TGA graphs indicated a weight loss at temperatures below 100 C, which might be due to the presence of residual solvent, which has not been accounted for.
  • A higher resolution image for Figure 4 should be provided.
  • The link for the optimized structures and interaction could not be accessed through the link http://doi.org/10.5281/zenodo.4147306.

Author Response

Referee 1

  1. Could the authors comment on the smaller size of modified silica particles compared to the unmodified ones, as shown in the SEM images, Figure 3.

Response: The un/functionalized flash stationary phases silica has been visualised with a 10 μm magnification whereases the un/functionalized HPLC stationary phase has a 1 um magnification.

  1. In the TGA analysis of the modified silica, Figure 2, the Cbutylpyrogallol[4]arene loading on the surface was determined by weight loss, however, the TGA graphs indicated a weight loss at temperatures below 100 C, which might be due to the presence of residual solvent, which has not been accounted for.

Response: The 4% weight loss at 100oC can be attributed to water, both on the surface of the silica and embedded within the host cavitand.

  1. A higher resolution image for Figure 4 should be provided.

Response: A new figure with high resolution has been added at the line 186.

  1. The link for the optimized structures and interaction could not be accessed through the link http://doi.org/10.5281/zenodo.4147306.

Response: There was an embargo access to the link, which has now been removed.

Reviewer 2 Report

1) In this paper authors describe the synthesis of the supramolecular cavitand C‐bromo‐butylpyrogallolarene 2 from C‐hydroxybutylpyrogallolarene. Have following minor questions:

2) How were C60 and C70‐fullerenes using the same chromatographic conditions chosen to validate stationary phase of hydroxybutylpyrogallolarene?

3) How significant is the retention time of fullerenes C60 and C70 compared to hydroxybutylpyrogallolarene?

4) Can authors label the atoms in Figure 6?

5) Many techniques/instrument versions are missing citations, please add them. Example: PuriFlash®5.125 flash system

6) Some references seem very old, if possible authors should use last 10 years of citations. Example,

Jinno, K.; Uemura, T.; Ohta, H.; Nagashima, H.; Itoh, K. Separation and Identification of Higher Molecular Weight Fullerenes 538 by High‐Performance Liquid Chromatography with Monomeric and Polymeric Octadecylsilica Bonded Phases. Anal. Chem. 539 1993, 65, 2650‐2654; Jinno, K.; Uemura, T.; Ohta, H.; Nagashima, H.; Itoh, K. Separation and Identification of Higher Molecular 540 Weight Fullerenes by High‐Performance Liquid Chromatography with Monomeric and Polymeric Octadecylsilica Bonded 541 Phases. Anal. Chem. 1993, 65, 2650‐2654

Author Response

Referee 2

In this paper authors describe the synthesis of the supramolecular cavitand C‐bromo‐butylpyrogallolarene 2 from C‐hydroxybutylpyrogallolarene. Have following minor questions:

  1. How were C60 and C70‐fullerenes using the same chromatographic conditions chosen to validate stationary phase of hydroxybutylpyrogallolarene?

Response: The same chromatographic conditions were chosen in order to evaluate the chromatographic performance of C60 and C70 fullerenes on C-butylpyrogallol[4]arene, based on in-silico modelling shows that the interaction of the fullerenes with the cavity increases with the size of the fullerene structure.

  1. How significant is the retention time of fullerenes C60 and C70 compared to hydroxybutylpyrogallolarene?

Response: The hydroxybutylpyrogallolarene was not bound to the surface of the silica, so we do not have a comparative study to compare to.

  1. Can authors label the atoms in Figure 6?

Response: In the line 280-281, the atoms  in Figure 6 has been added as “Where the red, light grey and dark grew spheres (or ribbon) represent oxygen, hydrogen, and carbon atoms respectively”.

  1. Many techniques/instrument versions are missing citations, please add them. Example: PuriFlash®5.125 flash system

Response: All instrumentation is fully referenced in section 3 Materials and Methods, 3.1 Reagents and instruments, lines 309-330

  1. Some references seem very old, if possible authors should use last 10 years of citations. Example, #Jinno, K.; Uemura, T.; Ohta, H.; Nagashima, H.; Itoh, K. Separation and Identification of Higher Molecular Weight Fullerenes 538 by High‐Performance Liquid Chromatography with Monomeric and Polymeric Octadecylsilica Bonded Phases. Anal. Chem. 539 1993, 65, 2650‐2654; Jinno, K.; Uemura, T.; Ohta, H.; Nagashima, H.; Itoh, K. Separation and Identification of Higher Molecular 540 Weight Fullerenes by High‐Performance Liquid Chromatography with Monomeric and Polymeric Octadecylsilica Bonded 541 Phases. Anal. Chem. 1993, 65, 2650‐2654

Response: The cited references are gold standard references to demonstrate the separation of fullerenes via octadecyl silica chromatographic phases.

Reviewer 3 Report

In this work, a C-butylpyrogallol[4] arene bonded silica is synthesized for column chromatography. When C-butylpyrogallol[4] arene bonded silica is used as the stationary phase, the HPLC results demonstrate better separation and resolution of C60 and C70 fullerenes than RP-C18. Density Functional Tight Binding (DFTB) simulations are performed to investigate the interaction between fullerenes and the stationary phase. I think this work merits to be published after the authors consider the following suggestions: 

  1. Some of the papers cited in this manuscript are not relevant to the discussions (such as the reference indicated in Line 42 and 43). The authors should check the references to make sure they are all correct.
  2. The definition of phase 3a and 3b needs to be clarified in the main text.
  3. In Table 1, HPLC and flash column have different pore volumes and sizes, but their surface area is identical. This is quite unusual so the data need to be double-checked.
  4. It is a good idea to carry out DFT calculations to investigate guest-host interactions in this work. However, since only the binding energies of C60 and C70 on butylpyrogallol[4]arene were calculated, these simulations can not explain why C-butylpyrogallol[4] arene bonded silica column shows a better separation of C60 and C70 fullerenes than RP-C18 (reference column). To shed light on this question, the binding energies of C60 and C70 on RP-C18 also need to be calculated.
  5. In Figure 4, broader peaks are obtained when C-butylpyrogallol[4] arene bonded silica column is used. The authors should discuss why tailing occurs in C-butylpyrogallol[4] arene bonded silica column and possible solutions to overcome this tailing effect.

Author Response

Referee 3

  1. Some of the papers cited in this manuscript are not relevant to the discussions (such as the reference indicated in Line 42 and 43). The authors should check the references to make sure they are all correct.

Response: Reference 9 has been removed.

2. The definition of phase 3a and 3b needs to be clarified in the main text.

Response: Phase 3a and phase 3b definition has addressed in the line 96-98 as “The silica-bound C- butylpyrogallol[4]arene HPLC stationary phase is reported as phase 3a while silica-bound C- butylpyrogallol[4]arene flash stationary phase is re-ported as phase 3b.

         3. In Table 1, HPLC and flash column have different pore volumes and sizes, but their surface area is identical. This is quite unusual so the data need to be double-checked.

Response: In table 1, The surface area for HPLC has been corrected and reported as 120 m2/g.

4. It is a good idea to carry out DFT calculations to investigate guest-host interactions in this work. However, since only the binding energies of C60 and C70 on butylpyrogallol[4]arene were calculated, these simulations can not explain why C-butylpyrogallol[4] arene bonded silica column shows a better separation of C60 and C70 fullerenes than RP-C18 (reference column). To shed light on this question, the binding energies of C60 and C70 on RP-C18 also need to be calculated.

Response: We have performed a DFTB calculation for C60⸦RP-C18 and C70⸦RP-C18. The details are found in the ESI. Also, we have added a paragraph in the main text (line 299 -307). These results show that C70 interacts more strongly with RP-C18 in comparison with C60, likely due to the higher curvature of C60. However, the fullerene⸦RP-C18 interactions are quite weak in comparison to the interactions between the fullerenes and the C-butylpyrogallol[4]arenes.

5, In Figure 4, broader peaks are obtained when C-butylpyrogallol[4] arene bonded silica column is used. The authors should discuss why tailing occurs in C-butylpyrogallol[4] arene bonded silica column and possible solutions to overcome this tailing effect.

Response: The conformational transformation of the host molecule, as it binds the guest, results in a kinetic distributional change that slows down the uptake and release of the fullerene guest and subsequent boarder peak shape. Selection of large pore size (300Å) chromatographic grade silica would minimize the peak tailing effect, while accounting sample and volume overloading which are responsible for peak tailing.